# A Survey on Companion Animal Owners’ Perception of Veterinarians’ Communication About Zoonoses and Antimicrobial Resistance in Germany

**DOI:** 10.3390/ani14223346

**Published:** 2024-11-20

**Authors:** Amelie Lisa Arnecke, Stefan Schwarz, Antina Lübke-Becker, Katharina Charlotte Jensen, Mahtab Bahramsoltani

**Affiliations:** 1Institute of Veterinary Anatomy, School of Veterinary Medicine, Freie Universität Berlin, Koserstraße 20, 14195 Berlin, Germany; al.arnecke@fu-berlin.de (A.L.A.); mahtab.bahramsoltani@fu-berlin.de (M.B.); 2Institute of Veterinary Pathology, Faculty of Veterinary Medicine, Leipzig University, An den Tierkliniken 33, 04103 Leipzig, Germany; 3Institute of Microbiology and Epizootics, School of Veterinary Medicine, Freie Universität Berlin, Robert-von-Ostertag-Straße 7, 14163 Berlin, Germany; antina.luebke-becker@fu-berlin.de; 4Veterinary Centre for Resistance Research (TZR), School of Veterinary Medicine, Freie Universität Berlin, Robert-von-Ostertag-Straße 8, 14163 Berlin, Germany; 5Institute for Veterinary Epidemiology and Biostatistics, School of Veterinary Medicine, Freie Universität Berlin, Königsweg 67, 14163 Berlin, Germany; charlotte.jensen@fu-berlin.de

**Keywords:** risk communication, risk perception, risk awareness, risk factors, zoonoses, antimicrobial resistance (AMR), pathogen transmission, one health, companion animals, online survey

## Abstract

Companion animals, such as dogs and cats, support the physical and psychological health of their owners. However, they can also carry pathogens that might be transmittable to humans. This risk is especially relevant for vulnerable groups, including young children, elderly, pregnant, and immunocompromised individuals. Therefore, risk communication by veterinarians is essential for raising awareness about associated risks. Our results showed that veterinarian risk communication was perceived well and rated with a high average score, indicating strong satisfaction among pet owners. Despite this positive feedback, there remained a gap between how often veterinarians provided information and the owners’ expressed need for further guidance. Thus, enhanced education and training of veterinary practitioners could narrow this gap. Overall, veterinarians play an important role in communicating risks associated with zoonoses and antimicrobial resistance, ultimately helping to improve the health of humans and animals.

## 1. Introduction

Companion animals, like dogs and cats, support physical health by encouraging exercise, improving cardiovascular health, boosting energy, and allowing their owners to maintain a healthy weight, while also promoting endorphin release. In addition, physical contact with pets boosts oxytocin levels, which strengthens bonding, reduces stress, and lowers cortisol levels, leading to relaxation [1,2,3,4]. Furthermore, findings suggest that pet ownership helps to reduce feelings of loneliness and its related effects, particularly in older adults living alone [5,6]. For children, bonding with pets can help in the development of empathy, social skills, emotional intelligence, and cognitive abilities, such as language and communication, while also teaching responsibility and nurturing behaviours [7,8]. Thus, pet ownership—especially of companion animals, such as dogs and cats—is common in Germany, particularly among families with children [9].

Despite all the benefits that pet animals bring to the lives of their owners, living with animals and having close contact also bears risks, such as the mutual transmission of pathogens through direct contact, fomite transmission, bites, or scratches [10,11]. There are multiple pathogens—e.g., parasites, such as *Giardia*, *Toxoplasma*, and *Toxocara*, bacteria, such as Enterobacterales, staphylococci, and pseudomonads, as well as fungi, such as dermatophytes—that can be transmitted from pets to their owners, potentially leading to illness [12,13,14,15,16,17,18,19,20,21,22,23,24,25,26,27,28,29,30,31,32,33].

However, zoonotic pathogens in companion animals are a growing concern in Germany, with only a few zoonoses (e.g., campylobacteriosis, echinococcosis, listeriosis, salmonellosis, SARS-CoV-2, tuberculosis and verotoxin-producing *Escherichia coli* infections) currently being considered as reportable animal diseases [34].

Furthermore, antimicrobial-resistant pathogens—including extended-spectrum beta-lactamase (ESBL)-producing Enterobacterales, such as *Escherichia coli* and *Klebsiella* spp., or methicillin-resistant *Staphylococcus aureus* (MRSA) and *Staphylococcus pseudintermedius* (MRSP)—pose a health threat, as they can reduce the effectiveness of treatment [35,36]. Since antimicrobial-resistant pathogens have been identified in companion animals, these animals may act as potential reservoirs [15,31,37,38]. This increases the likelihood of these antimicrobial-resistant pathogens being transferred between pets and their owners [39,40,41,42,43]. Antimicrobial resistance (AMR)—also referred to as the ‘silent pandemic’—is becoming increasingly relevant due to the rising use of antibacterial cleaning products and medical treatments [44]. Given the close interaction between humans and their animals, it is important to recognise the various pathways of transmission, especially in relation to individual risks faced by vulnerable groups, such as children under 6 adults over 65, and immunocompromised and pregnant individuals [11,45,46,47,48,49,50,51]. Therefore, maintaining proper hygiene and possessing a certain level of knowledge and risk awareness are crucial for realistically assessing associated risks and implementing individually tailored protective measures, including preventive healthcare, such as vaccination and the use of antiparasitics [46,52,53,54,55,56,57].

Risk communication is an important aspect of health education, as it continuously and preventively conveys knowledge to establish risk awareness in the general population [58,59,60,61]. In addition, medical progress, improvements in living conditions and hygiene, as well as education and the targeted use of measures against communicable diseases, have led to a significant decline in infectious diseases in the 20th century [62]. Thus, health education, science communication, and knowledge transfer represent essential methods for preventing disease outbreaks and their spread by creating risk awareness in the community and increasing participation in preventive behaviour [63,64,65,66,67].

Medical professionals play a crucial role in translating risk communication into clinical practice [68]. Effective communication is a key factor for patient satisfaction, compliance, and recovery [69]. Good communication skills are a necessary and important qualification in healthcare, contributing to successful collaboration between healthcare professionals and their clients, leading to positive treatment outcomes [70]. Beyond the quality of care, other important skills that contribute to customer satisfaction include the transfer of information, respect, honesty, transparency, and empathy [71]. Similarly, veterinarians play an important role in informing and educating pet owners about health risks related to zoonoses and AMR [48,72,73,74,75,76]. Thus, veterinarians should have good communication skills to effectively transfer information, aiming to create risk awareness among pet owners. Therefore, veterinary consultations could help improve the targeted dissemination of information and education in the future, such as by informing pet owners of signs and symptoms of illness to prevent pathogen transmission [77,78]. However, a close and trusting relationship is primarily based on communication that is clear and understandable [79], and the explanation of various preventive measures as well as the communication style are key factors in this regard [80]. Despite this, veterinarians often face challenges in interacting and communicating effectively with pet owners due to the involvement of stress and emotions [81,82]. In addition, the perception of risk varies depending on factors, such as expectations, personal needs, and past experiences [83]. Therefore, understanding the perception of veterinarian communication is essential for adapting risk communication strategies [24,84].

Hence, this study aimed to explore veterinarian communication on zoonoses and AMR, particularly in relation to risk factors and risk behaviour. Furthermore, companion pet owners’ perception of veterinarian communication was assessed. The findings provide important insights for practicing veterinarians, helping them to adapt communication strategies aimed at strengthening the health of both humans and animals.

## 2. Participants, Materials, Methods

### 2.1. Participants and Materials

This study targeted companion animal owners (i.e., owners of dogs and cats) in Germany, with data collected through an online survey which was conducted via LimeSurvey Cloud Version 5.6.56^®^ between December 2022 and April 2023. The survey was promoted on social media platforms, including Facebook and Instagram, and participants accessed it by scanning a QR code or following a survey link. Before beginning the survey, participants were required to read and agree to the legal notice and privacy policy, as well as confirm that they were at least 18 years old [73].

### 2.2. Questionnaire

The questionnaire was divided into six sections. Participants were asked about vulnerable groups, their pets and related risk behaviour, the veterinarian–client relationship, and whether they had visited a veterinarian within the preceding 12 months. We made a distinction between routine visits (such as check-ups for disease prevention, vaccination, blood tests, and dental care) and consultations due to a specific reason such as clinical symptoms, such as coughing or diarrhoea, and/or the detection of a pathogen. If participants indicated that they had consulted a veterinarian, they were further questioned on whether communication about zoonoses and/or AMR occurred. If communication took place, they were asked to assess their perception of the information received. If not, they proceeded to the final section, which covered veterinary advice and unmet needs. To minimise misunderstandings, technical terms were explained beforehand and reiterated in the relevant sections [73]. The definitions of zoonoses and AMR that were used were based on those provided by the World Health Organization [35,85]. Some questions were only asked if certain requirements were met, and not all questions were mandatory. Therefore, the number of answers differed [73].

#### 2.2.1. Vulnerable Individuals

The first section of the questionnaire collected data on vulnerable individuals within the participants’ households, including pregnant people, children under 6 years of age, elderly individuals over 65 years of age, and vulnerability connected to health-related factors (Table 1). The category ‘children’ was created by merging ‘infant’ and ‘toddler’. Similarly, the category ‘health-related’ was formed by combining the variables ‘chronically ill’, ‘diabetes’, ‘cancer’, and ‘other conditions that lead to immunosuppression’, such as medical treatments like chemotherapy and radiation therapy [73].

#### 2.2.2. Pets, Risk Behaviour, Risk Awareness

The second section gathered information on the respondents’ pets and associated risk behaviour connected to the pets’ origin (imported from abroad), and the respondents’ involvement in risk-associated feeding practices (raw meat/fish, fresh offal, and uncooked bones) [73]. Furthermore, the awareness of risks connected to risk-associated feeding practices was assessed (Table 2).

#### 2.2.3. Veterinarian–Client Relationship

The third section evaluated the permanence and duration of the respondents’ relationships with their veterinarians, as well as their level of satisfaction with the care provided (Table 3) [73].

#### 2.2.4. Veterinary Consultation

The section on veterinary consultation assessed whether participants had consulted a veterinarian within the preceding 12 months and the reason for consultation (Table 4). Consultations were categorised into routine visits, which included preventive check-ups, such as consultations due to vaccinations, testing of blood and serum, or examinations of teeth, and consultations due to specific symptoms and/or the detection of a pathogen. The variable named ‘specific’ was created by combining the various symptom-related variables, such as gastrointestinal, skin, urinary, respiratory, and others. Pathogen detection was assessed by using a list of the most relevant pathogens (Appendix A). This list was compiled through a literature review and consultations with diagnosticians and practicing veterinarians [73].

#### 2.2.5. Veterinary Communication

##### Frequency and Topics of Communication

If the participants indicated that they had visited a veterinarian for a specific or a routine reason, they were questioned whether the veterinarian had provided them with information about zoonoses and/or AMR [73]. Furthermore, if they stated to have consulted a veterinarian due to a routine visit, they were asked, which topics the veterinarian talked about (Table 5). The variable ‘hygiene measures’ was created by combining the variables ‘picking up dog waste’, ‘hand hygiene’, ‘cleaning of food and water bowls’, ‘washing textiles’, ‘associated risks for pregnant people when cleaning the cat litter’, ‘birth hygiene’, ‘covering of children’s sandpit’, ‘cleaning vegetables from the garden’ and ‘other’.

##### Perception of Veterinarian Communication

To assess the pet owners’ perception of their veterinarian, a self-developed and validated questionnaire was used [73]. The questionnaire items were formulated based on Friedemann Schulz von Thun’s four-sides model, which suggests that every message contains four dimensions: factual, appeal, relationship, and self-disclosure. The factual items assessed whether the information was relevant and understandable to the pet owner. The appeal level items measured whether requests were clearly communicated and whether the consequences of following or not following the advice were explained. The relationship level items focused on the pet owners’ perception of their veterinarians’ trustworthiness and empathy. Finally, the self-disclosure items aimed to determine if veterinarians’ self-presentation conveyed professional competence [86]. Based on this model, four items were formulated for each dimension, with each item rated by participants on a 4-point Likert scale: 1 = disagree, 2 = rather disagree, 3 = rather agree, and 4 = agree, with an additional "no answer" option (Table 6) [73].

A factor analysis was conducted with four items for each dimension to evaluate the validity of the questionnaire. The goal was to determine if these four dimensions could effectively assess the different levels of communication [73]. A Cronbach’s Alpha of 0.891 indicated high internal consistency across the 16 items. Thus, sensitivity controls showed that the sides were not distinct. In addition, exploratory factor analysis failed to separate the sides into distinct factors. Bartlett’s test of sphericity (*p* < 0.001) suggested high correlations among the variables of communication, and the Kaiser–Meyer–Olkin measure of sampling adequacy was 0.567. As a result, the mean value—a one-dimensional score—was calculated and used to represent perceived communication [73].

#### 2.2.6. Veterinary Advice and Needs of Companion Pet Owners

The last section explored respondents’ concluding thoughts on veterinary advice, assessing their interest in and need for guidance in various areas, including zoonoses, AMR, behaviour, nutrition, vaccination, animal husbandry, hygiene measures, and the treatment of endo- and ectoparasites (Table 7) [73].

### 2.3. Statistical Analysis

The data were exported to Microsoft Excel 2024^®^. Statistical analyses were performed using IBM SPSS Statistics Version 29^®^. Only participants who completed all sections of the questionnaire were included in the analyses. However, since no questions were mandatory, participants had the option to skip individual questions within each section [73].

#### 2.3.1. Analysis of Communication Frequencies

To understand risk communication related to zoonoses and AMR, an investigation was performed to determine whether such communication takes place and if risk factors—such as the presence of vulnerable individuals in the household and the detection of pathogens—along with risk-associated behaviours (e.g., feeding raw meat or fish, offering fresh offal or uncooked bones, and/or owning an animal imported from abroad) were associated with the frequency of communication. A Chi^2^-Test was utilised to analyse the association between two categorical variables; if the expected count in any cell was less than five, Fisher’s exact test was applied instead [73].

#### 2.3.2. Analysis of Pet Owners’ Communication Perception

The scores of pet owners’ perception of communication were compared based on the presence or absence of risk factors (vulnerable groups, pathogen detection), risk behaviours (risk-associated feeding practices, imported animals), and the reason for consultation (specific vs. routine). Since the data were normally distributed, the independent samples *t*-test was used to compare the two groups. Furthermore, this analysis aggregated cases of confirmed pathogens without distinguishing between those with zoonotic potential and those with AMR. Additionally, for the duration of veterinary care, a factorial analysis of variance was used to compare more than two groups (under one year, one to three years, three to six years, over six years). A significance level of 5% was set for all inferential analyses [73].

## 3. Results

A total of 1315 people participated in the study, of whom 1015 completed the survey. As some questions were only asked if certain requirements were met and not all questions were mandatory, the number of answers differs in the following sections. Of those who completed the survey, 67% (n = 674 of 1010) reported owning dogs and 53% (n = 536 of 1007) reported owning cats.

### 3.1. The Presence of Vulnerable Groups in the Household

About 43% of the participants (n = 437 of 1015) reported having individuals of vulnerable groups in their households, including people with health-related vulnerabilities—such as chronic illnesses, cancer, diabetes, or other conditions that lead to immunosuppression—followed by elderly individuals over 65 years of age, children under 6 years of age, and pregnant individuals (Figure 1).

### 3.2. Pets, Risk Behaviour, Risk Awareness

About 44% (n = 442 of 1015) of the participants reported engaging in activities related to risk behaviour including owning an imported animal (27%, n = 269 of 990) and risk-associated feeding practices, such as the feeding of raw meat or fish, fresh offal, and uncooked bones (23%, n = 233 of 1002).

About 67% (n = 578 of 866) of the participants were aware of the risks connected to risk-associated feeding practices, and the majority who answered the question (n = 578 of 866) associated the risk with the possibility of pathogen transmission in addition to nutrient deficiency and intolerance (Figure 2).

In the group that was involved in risk-associated feeding practices and answered the question (n = 222 of 233), 45% (n = 99 of 222) were aware, whereas 55% (n = 123 of 222) were not aware of associated risks. However, in the group that was not involved in these feeding practices, 74% (n = 475 of 639) were aware and 26% (n = 164 of 639) were not aware of associated risks. This finding was significant (*p* < 0.001).

### 3.3. Permanence, Satisfaction with, and Duration of Veterinary Care

Among the participants, 94% (n = 947 of 1005) reported having a regular veterinarian, with the majority having long-lasting relationships. Furthermore, of the participants who answered the question (n = 944 of 947), 73% (n = 687 of 944) reported a high level of customer satisfaction (Figure 3).

The two main reasons for participants not having a permanent veterinarian (6%, n = 58 of 1005) were that the clients either consulted multiple veterinarians or were dissatisfied with the quality of care. Additional reasons included no perceived need and the distance from available veterinarians (Figure 4).

### 3.4. Reasons for Veterinary Consultations

A high number of the participants (95%; n = 957 of 1012) stated that they had consulted a veterinarian within the preceding 12 months. Among those consultations, 74% (n = 747 of 1012) involved a routine visit and 42% (n = 430 of 1012) were because of a specific symptom of their pet. For participants who stated that they consulted a veterinarian due to specific symptoms, more than half of the animals had gastrointestinal symptoms, followed by skin, respiratory, and urinary symptoms (Figure 5).

Of participants who reported that they consulted a veterinarian within the preceding 12 months, 10% (n = 93 of 957) reported pathogen detection with zoonotic potential (9%, n = 85) and/or AMR (1%, n = 13). A total of 114 cases, including bacterial, protozoan, fungal, and endo- and ectoparasite pathogens, were reported. *Giardia*, dermatophytes, and *Escherichia coli* were the most frequently detected pathogens (Figure 6).

### 3.5. Veterinarian Communication

#### 3.5.1. Frequency of Veterinary Communication

Among the participants who visited a veterinarian within the preceding 12 months, 33% (n = 316 of 957) reported that they received information, while 29% reported receiving information on zoonoses (n = 275 of 957) and 12% on AMR (n = 116 of 957). The communication frequency was associated with the permanence of veterinarian supervision (*p* = 0.003). The longer the pet owners were clients, the more frequently they received information on zoonoses and AMR (<1 year: 29%, n = 30 of 103; 1–3 years: 30%, n = 85 of 288; 3–6 years: 33%, n = 65 of 198; >6 years: 35%, n = 124 of 353). This result was not statistically significant (*p* = 0.429).

An association between communication and the existence of vulnerable individuals in the participants’ households (*p* = 0.024) was detected. There were no associations with risk behaviours (*p* = 0.264). However, an association was detected between communication frequency (n = 291 of 866) and risk awareness (n = 578 of 866) related to risk-associated feeding practices (*p* < 0.001). If participants reported being aware of risk-associated feeding practices, a higher frequency of communication was observed (38%, n = 221 of 578) compared to when participants were not aware of associated risks (24%, n = 70 of 288).

#### 3.5.2. Topics of Veterinary Communication

Participants who visited the veterinarian for a routine check-up received information on vaccination, the treatment of endo- and ectoparasites, hygiene measures, the handling of animal bites, risk-associated feeding practices, and other topics (Figure 7).

#### 3.5.3. Pet Owners’ Perception of Veterinarian Communication

The perception of veterinarian communication received, in general, a high average score of 3.4 out of a maximum of 4.0 (mean; n = 283). Most participants responded with ‘agree’ or ‘rather agree’ (Figure 8). Differences were apparent especially concerning the first three items of the appeal level, where consultations due to specific reasons received a higher percentage of disagreement. Furthermore, item No. 4 (‘The veterinarian provided me with information material.’) of the appeal level received the highest disagreement score. During visits due to routine reasons, information material was distributed in 39% of cases, whereas during consultations due to specific reasons, information material was distributed in 21% of cases.

##### Communication Perception Connected to Risk Factors and Risk Behaviour

The presence and non-presence of vulnerable individuals in the participants’ household resulted in the same average score of 3.4 (yes: n = 144; no: n = 140). Similar to vulnerable groups, the presence and non-presence of behaviour associated with risks resulted in the same average score of 3.4 (yes: n = 124; no: n = 160). If a pathogen was detected, the average communication score (mean = 3.3, n = 124) was significantly lower (*p* < 0.001) in comparison to communication when no pathogen was detected (mean = 3.5, n = 160).

##### Communication Perception Connected to Veterinary Care Duration

Among the participants with a permanent veterinarian, there was a tendency that the longer the pet owners were clients, the better they rated the communication. This result was not statistically significant (*p* = 0.559) (Figure 9).

##### Communication Perception Connected to the Reason for Consultation

Communication perception during a routine check-up consultation had a higher average communication score of 3.5 (n = 207) than for consultations due to specific reasons (mean = 3.2, n = 76), with a statistically significant difference (*p* < 0.001) (Figure 10).

### 3.6. Veterinary Advice and Needs

Most participants expressed interest in receiving veterinary advice on zoonoses (68%, n = 626 of 917) and AMR (72%, n = 659 of 918). In addition, pet owners wanted veterinarians to offer information on nutrition, the treatment of endoparasites, behaviour, vaccination, the treatment of ectoparasites, hygiene measures, animal husbandry, and other related topics (Figure 11).

## 4. Discussion

Pet ownership can result in the transmission of zoonotic and antimicrobial-resistant pathogens between pets and humans in either direction [11,12,13,14,15,16,17,18,19,20,21,40,41,42,43]. It is a key responsibility of veterinarians to inform pet owners about these risks [48,72,73,74,75,87]. The effectiveness of this information exchange largely depends on veterinarians’ knowledge and communication skills, to ensure that pet owners understand the risks and follow the provided guidance [79,80,81,82,83,84]. Thus, this study aimed to evaluate how veterinarians address the risks of zoonoses and AMR in their communication and to examine pet owners’ perceptions of their veterinarians’ communication on these topics.

A significant proportion of participants (95%) consulted a veterinarian within the preceding 12 months, with the majority (74%) doing so for routine visits or preventive check-ups. This high rate of veterinary consultation is consistent with the general opinion that preventive care is considered the foundation of animal health care and protection from the emergence of diseases [88,89]. Participants who sought veterinary care because of specific symptoms of their pets (42%) most commonly reported concerns connected to the gastrointestinal tract (56%) followed by the skin (36%). These findings align well with previous studies that investigated primary routes of transmission for zoonoses and AMR [90,91,92,93].

Previous studies have identified potentially zoonotic and antimicrobial-resistant pathogens in companion animals, indicating a clear risk of transmission to humans [11,15,16,17,18,19,20,21,22,31,37,38,94,95]. In this study, 10% of the companion pet owners who had visited a veterinarian within the preceding 12 months reported pathogen detection in their pets, with zoonotic potential (9%) or/and AMR (1%). Altogether, 114 cases were documented, including bacteria, protozoa, fungi, and endo- and ectoparasites. *Giardia*, dermatophytes, and *E. coli* were the pathogens detected most frequently. The presence of these pathogens highlights the potential health risks pets may pose to humans [23,24,26,27,96], particularly regarding gastrointestinal symptoms, such as diarrhoea, dehydration, gas, and stomach cramps [9,97,98]. It also underscores the risk of urinary tract infections, which can cause painful urination, urine odour, fever, and frequent urination [99], as well as skin-related symptoms, such as lesions, inflammatory nodules, and pustules [100,101]. Furthermore, this also increases pathogen shedding from pets and the likelihood of transmission to humans, either through direct contact with the animal itself or indirectly via indoor soiling and the contamination of shared living spaces [47].

The results of this study showed that 43% of the participants also have members of vulnerable groups in their households. Studies have shown that children under 6, adults over 65, people with health conditions, and pregnant individuals are at higher risk of becoming ill and may experience more severe symptoms or faster disease progression than the general population [45,46,47,48,49,50,51,72].

Furthermore, 44% of the respondents indicated engagement in activities that could increase the risk of pathogen transmission, such as the feeding of raw meat or fish, fresh offal, and uncooked bones and owning animals that were imported from other countries to Germany. Research on raw meat feeding practices has confirmed the presence of pathogens, including antimicrobial-resistant strains, that can lead to serious illnesses in humans. Pets consuming raw meat can carry these pathogens asymptomatically, acting as reservoirs and potential sources of transmission [102,103]. This poses a particularly high risk to vulnerable groups, who are more susceptible to severe illness when handling such products [104,105,106]. Most of the participants (67%) indicated an awareness of risks associated with these feeding practices, with the majority associating these risks with potential pathogen transmission. This high level of awareness could potentially be linked to the various campaigns during the COVID-19 pandemic, which were aimed at educating the public about the dangers of zoonoses [61,107,108]. However, awareness did not necessarily translate into action. In the group that reported engaging in risk-associated feeding practices, only 43% were aware of associated risks. This might be due to the underestimation of potential health risks [109]. However, 55% of the participants that stated that they engaged in risk-associated feeding practices were unaware of the potential health risks. This highlights a significant gap in risk awareness, which could lead to the spread of zoonotic and antimicrobial-resistant pathogens, especially given the possibility of asymptomatic carriage of these pathogens by animals consuming raw meat [110,111,112].

Furthermore, 27% of the participants stated that they owned animals that were imported to Germany. Depending on their origin, animals may pose a risk to the health of humans and other animals due to their potential to carry microorganisms that cause diseases like rabies, echinococcosis, and leishmaniosis [113]. However, imported animals can also carry *Giardia*, dermatophytes, and *E. coli*, as these pathogens have higher prevalences in shelter dogs in comparison to family dogs [28,114,115,116] and have a high prevalence in south-eastern European countries with populations of free-roaming dogs and cats [114,117,118,119,120,121,122,123]. Given that some of these pathogens may be asymptomatic in animals, it is essential to implement and follow health protocols, including health check-ups and vaccination, before the import process to prevent the introduction and spread of the respective infectious diseases [29,30,56,124,125,126].

As risk-associated feeding practices and import from other countries increase the likelihood of zoonotic transmission and other health risks for both animals and humans, there is a need for targeted risk communication strategies connected to individual risk profiles. Preventive measures such as regular deworming, vaccinations, or screening for diseases should be emphasised, especially in households with vulnerable individuals [56,57,127]. Therefore, tailored risk communication from veterinarians could be a tool for raising awareness and potentially preventing pathogen transmission [48,72,73,74,75,128,129,130].

A notable 94% of participants reported having a regular veterinarian, with most maintaining long-term relationships. This trend suggests a strong, trusted bond between pet owners and veterinarians, reflected further by the high customer satisfaction rate of 73%. Long-standing relationships foster trust and facilitate better communication, which are crucial for effective medical care [131,132]. Communication about zoonoses and AMR was significantly more frequent among pet owners who had a permanent veterinarian. Moreover, longer client–veterinarian relationships were associated with increased communication on these topics, indicating the importance of continuous care. These findings are consistent with previous research, further emphasising the crucial role of the veterinarian–client relationship in effective communication [73,79]. A strong relationship between veterinarians and pet owners has been shown to enhance compliance with veterinary recommendations, leading to improved health outcomes [71,73,133,134].

Among those participants who visited the veterinarian within the preceding 12 months, 29% reported receiving information on zoonoses and 12% on AMR. Furthermore, information material was distributed in 39% of cases during routine consultations, but only in 21% of the consultations due to specific reasons.

In addition, participants with vulnerable individuals in their households reported more frequent communication, aligning with the higher need for tailored advice to mitigate risks. Participants who were aware of risks connected to risk-associated feeding practices also reported being given information by their veterinarians more frequently, possibly indicating an active engagement in asking for information. Moreover, when communication on zoonoses and AMR occurred, veterinary communication was perceived very positively, receiving a high average score and indicating a high level of satisfaction. However, the presence of vulnerable individuals or risk behaviour did not significantly impact communication scores, suggesting a consistent communication quality regardless of risks.

The average perception of communication score was 3.4 out of 4, with routine visits receiving a significantly higher score (3.5) compared to consultations for specific reasons (3.2). This difference suggests that routine visits provide a more conducive environment for effective communication, possibly due to less stress compared to visits triggered by acute health concerns [135,136].

Lastly, participants expressed a desire for more information on zoonoses (68%) and AMR (72%), in addition to topics such as nutrition, endoparasites, and hygiene measures. Although veterinarians had already provided guidance on vaccination and the treatment of parasites, these results indicated that there is still a gap in addressing needs connected to information on zoonotic diseases and AMR. By expanding communication on these topics, veterinarians might better meet the needs of pet owners.

There were several limitations related to participant engagement in this study. The online distribution method might have created a barrier for some individuals, likely leading to a selection bias. Consequently, the participants may not fully represent the target population, since participation demands time, willingness to engage in surveys, and an interest in the subject of the survey. Although the recall period was limited to 12 months, there might be a memory bias, as respondents may inaccurately remember or base their responses on more recent or emotionally charged experiences, potentially affecting the reliability and validity of the data. Furthermore, these answers were self-declared data from pet owners and had several limitations including subjectivity and recall bias, as pet owners might overestimate their care practices or inaccurately remember details [137]. In addition, veterinary consultations could be stressful, meaning that the information conveyed by the veterinarian might not always be fully understood by the pet owner [135,136]. Another limitation was the use of the 4-point Likert scale in the section on communication perception, which might have offered too few response options to accurately reflect nuanced perceptions. Another limitation is the lack of geographical data within Germany, which prevents analysis of regional differences in pet owner participation. In future surveys, adding location-specific questions could allow insights into regional trends in pet ownership and participation rates. Additionally, expanding the survey to include other regions and countries would provide a broader understanding of pet ownership patterns on an international scale.

While this study emphasises the crucial role of veterinarians in risk communication during consultations, we see strong potential in broadening these efforts. Future perspectives should consider expanding awareness initiatives (e.g., workshops, lectures, courses, and seminars) for pet owners, addressing signs and symptoms of illness but also individual risk factors [138,139,140]. With this, owners could be empowered to take proactive health measures. Starting with information shared in clinics and scaling up to community-level campaigns would likely improve public understanding and proactive health measures [108,141].

## 5. Conclusions

The results of this study underscore the critical need for enhanced risk communication and education efforts regarding zoonoses, AMR, and responsible pet ownership. Risk-specific messaging is essential, particularly in the form of targeted recommendations for households with imported animals, risk-associated feeding practices, and vulnerable individuals. Furthermore, the findings emphasise the importance of routine veterinary care, including deworming, vaccination, and hygiene measures for protecting pets and humans from zoonotic diseases and AMR, as numerous pathogens were detected. Therefore, veterinarians play an important role in communicating health risks associated with zoonoses and AMR. The permanence and duration of the veterinarian–client relationship contributes to a higher frequency of information dissemination and enhances pet owners’ perception of veterinarian communication—reflected by the high level of satisfaction among pet owners. Thus, taking an active role in bridging the gap between risk awareness and behaviour by providing more information for safer practices might be effective. Consequently, public health and veterinary authorities should develop targeted communication strategies to mitigate the risks associated with pet ownership, while also enhancing the education and training of veterinary practitioners.

## Figures and Tables

**Figure 1 animals-14-03346-f001:**
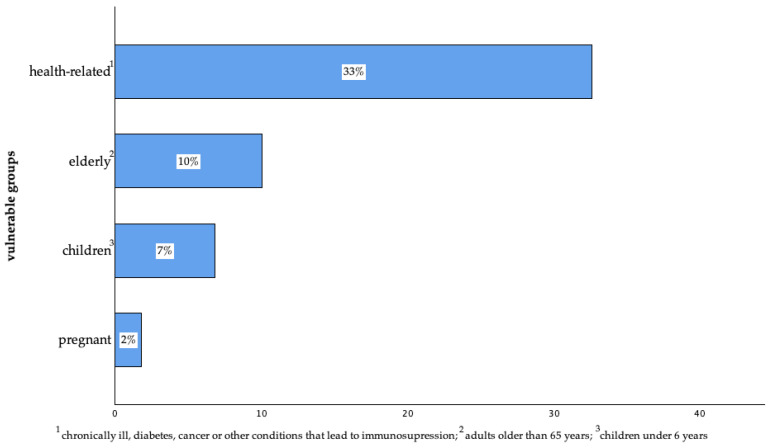
The presence of vulnerable groups in the household of surveyed pet owners in Germany. (Health-related: n = 331; elderly: n = 102; children: n = 69; pregnant: n = 18 of 1015; multiple answers were possible).

**Figure 2 animals-14-03346-f002:**
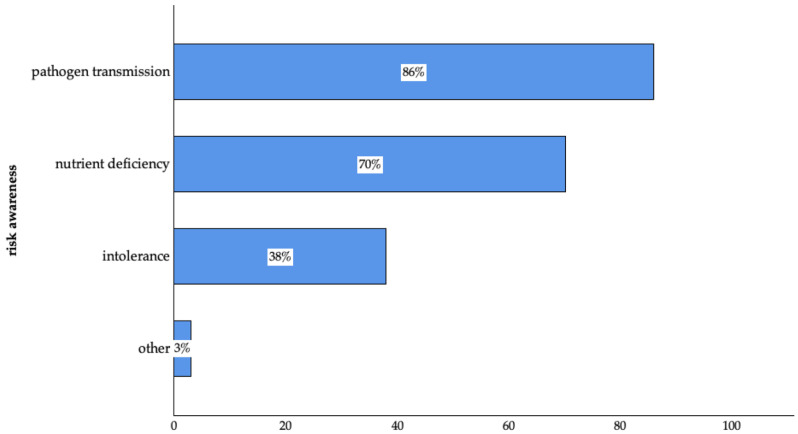
Risk awareness connected to risk-associated feeding practices of surveyed pet owners in Germany. (Pathogen transmission: n = 497; nutrient deficiency: n = 406; intolerance: n = 220; other: n = 15 of 578; multiple answers were possible).

**Figure 3 animals-14-03346-f003:**
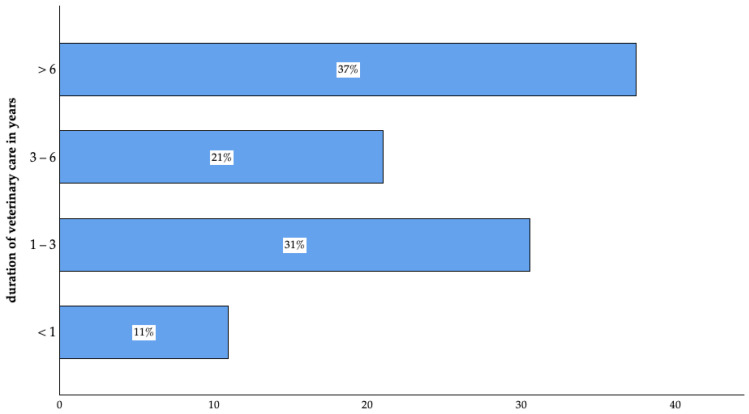
Duration of veterinary care of surveyed pet owners in Germany (<1 year: n = 103; 1–3 years: n = 288; 3–6 years: n = 198; >6 years: n = 353 of 942).

**Figure 4 animals-14-03346-f004:**
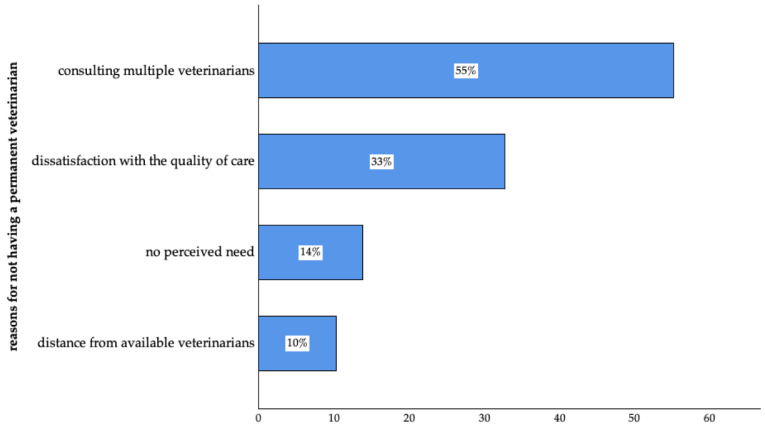
Reasons for not having a permanent veterinarian of surveyed pet owners in Germany. (Consulting multiple veterinarians: n = 32; dissatisfaction with the quality of care: n = 19; no perceived need: n = 8; distance from available veterinarian: n = 6 of 58; multiple answers were possible).

**Figure 5 animals-14-03346-f005:**
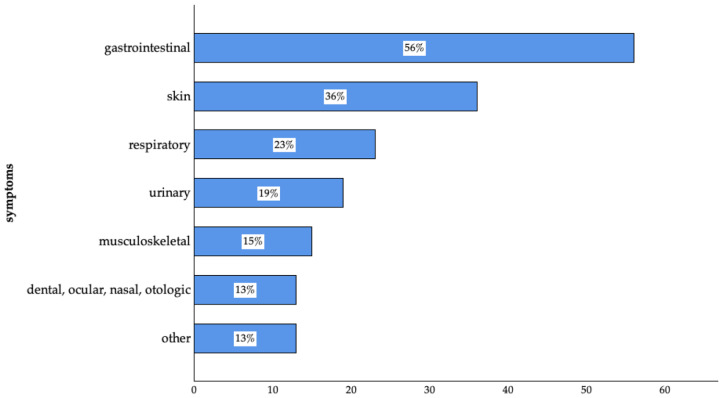
Symptoms reported by surveyed pet owners in Germany. (Gastrointestinal: n = 241; skin: n = 155; respiratory: n = 101; urinary: n = 82; musculoskeletal: n = 64; dental, ocular, nasal, otologic: n = 58; others: n = 55 of 430; multiple answers were possible).

**Figure 6 animals-14-03346-f006:**
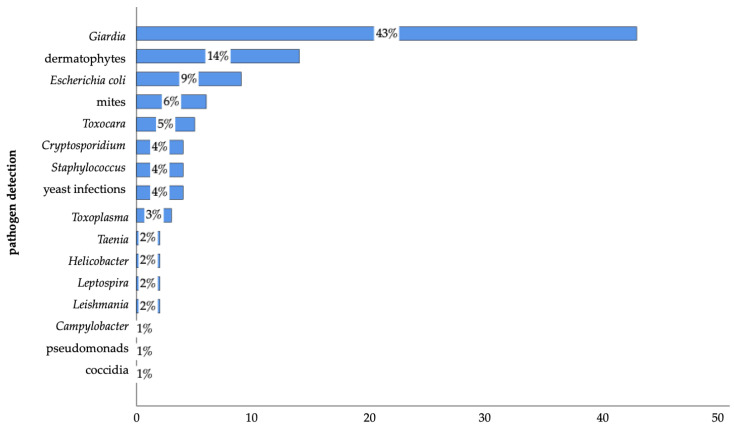
Pathogen detection reported by surveyed pet owners in Germany. (*Giardia*: n = 49; dermatophytes: n = 16; *Escherichia coli*: n = 10; mites: n = 7; *Toxocara*: n = 6; *Cryptosporidium*: n = 5; *Staphylococcus*: n = 4; yeast infections: n = 4; *Toxoplasma*: n = 3; *Taenia*: n = 2; *Helicobacter*: n = 2; *Leptospira*: n = 2; *Leishmania*: n = 1; *Campylobacter*: n = 1; pseudomonads: n = 1, coccidia: n = 1 of 114; multiple answers were possible).

**Figure 7 animals-14-03346-f007:**
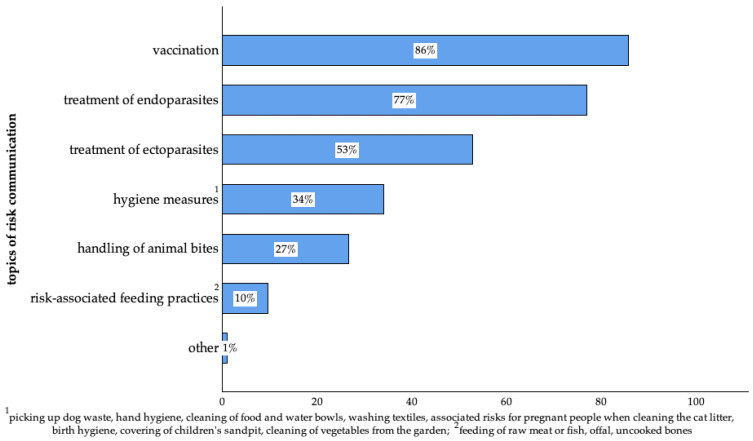
Topics of communication during routine visits mentioned by surveyed pet owners in Germany. (Vaccination: n = 640; treatment of endoparasites: n = 574; treatment of ectoparasites: n = 394; hygiene measures: n = 251; handling of animal bites: n = 199; risk-associated feeding practices: n = 72; other: n = 6 of 747; multiple answers were possible).

**Figure 8 animals-14-03346-f008:**
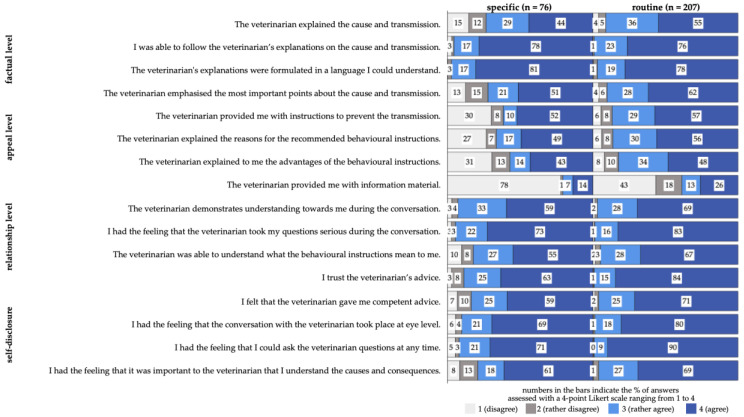
Communication perception of surveyed pet owners in Germany connected to Schulz von Thun’s four-sides model. The four levels of communication (factual level, appeal level, relationship level, self-disclosure) were used to formulate question items for each level of communication. It was assessed on a 4-point Likert scale ranging from 1 to 4: 1 (disagree), 2 (rather disagree), 3 (rather agree), 4 (agree).

**Figure 9 animals-14-03346-f009:**
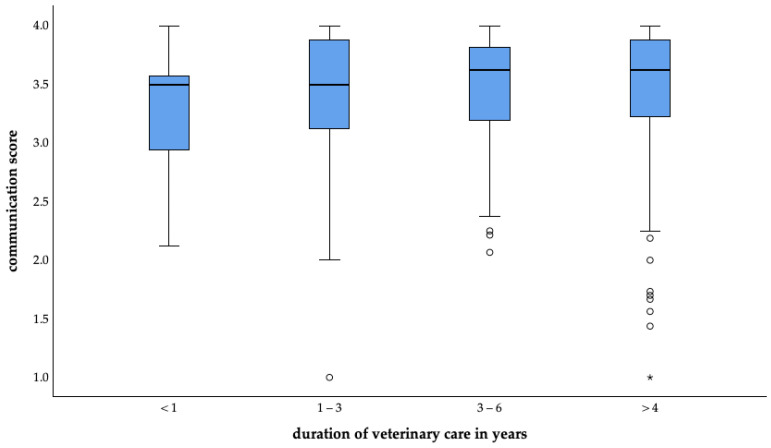
Average communication perception of surveyed pet owners in Germany connected to veterinary care duration. (Under one year: mean = 3.3, n = 21; one to two years: mean = 3.4, n = 61; three to six years: mean = 3.5, n = 64; over six years: mean = 3.5, n = 124; o = outlier, * = extreme value).

**Figure 10 animals-14-03346-f010:**
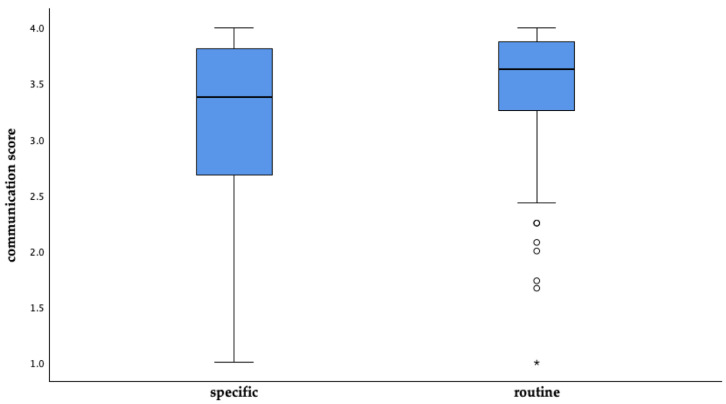
Average communication perception of surveyed pet owners in Germany in relation to the reason for consultation. (Specific: mean = 3.2, n = 76; routine: mean 3.5, n = 207; o = outlier, * = extreme value).

**Figure 11 animals-14-03346-f011:**
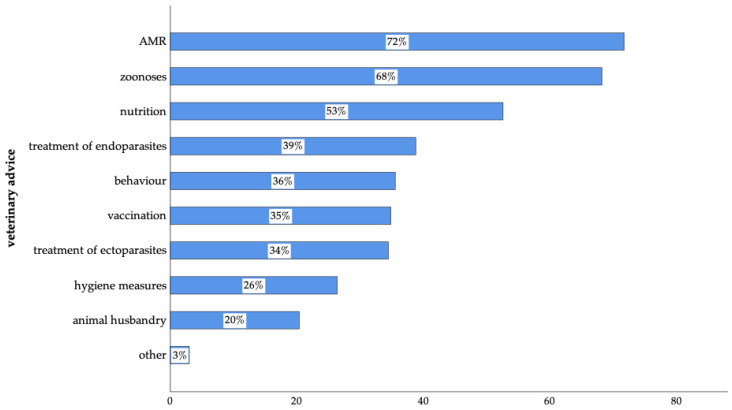
Topics of veterinary advice and needs of surveyed pet owners in Germany (AMR: n = 659 of 918; zoonoses: n = 626 of 917; nutrition: n = 534; treatment of endoparasites: n = 394; behaviour: n = 361; vaccination: n = 354; treatment of ectoparasites: n = 350; hygiene measures: n = 268; animal husbandry: n = 207; other: n = 27 of 1015; multiple answers were possible).

**Table 1 animals-14-03346-t001:** Questions: vulnerable individuals. (multiple answers were possible: 1. and 2.) [73].

1. Which age groups of people live in your household? (infants ^1^, toddlers ^2^, school-aged children, teenagers, adults, elderly ^3^) 2. Do you have individuals of the following groups in your household? (pregnant, chronically ill, diabetes, cancer, other conditions that lead to immunosuppression ^4^)

^1^ Under 2 years, ^2^ 2 to 5 years, ^3^ over 65 years, ^4^ congenital or acquired (e.g., medical treatments like chemotherapy and radiation therapy).

**Table 2 animals-14-03346-t002:** Questions: pets, risk behaviour, risk awareness. (multiple answers were possible: 6.) [73].

1. Do you have a dog? (yes, no) 2. Do you have a cat? (yes, no) 3. Is your pet imported from abroad? (yes, no) 4. Do you feed raw meat/fish, fresh offal, uncooked bones? (yes, no) 5. Do you see a risk connected to feeding raw meat/fish, fresh offal, uncooked bones? (yes, no) if yes: 6. Which risks do you see connected to the feeding of raw meat/fish, fresh offal, uncooked bones? (pathogen transmission, nutrient deficiency, intolerance, other)

**Table 3 animals-14-03346-t003:** Questions: veterinarian–client relationship. (multiple answers were possible: 4.) [73].

1. Do you have a permanent veterinarian? (yes, no) if yes: 2. How long have you been a regular client of your veterinarian? (<1 year, 1–3 years, 3–6 years, >6 years) 3. Are you satisfied with the care you receive from your veterinarian? (disagree, rather disagree, rather agree, agree) if no: 4. What is the reason for not having a permanent veterinarian? (consulting multiple veterinarians, dissatisfaction with the quality of care, no perceived need, distance from available veterinarian)

**Table 4 animals-14-03346-t004:** Questions: veterinary consultation. (multiple answers were possible: 2., 3., 5.) [73].

1. Have you visited a veterinarian within the past 12 months? (yes, no) if yes: 2. For which reason have you consulted the veterinarian? (routine check-up, acute or chronic illness, emergency, surgery, other) 3. Does or did your pet have symptoms? (gastrointestinal, skin, respiratory, urinary, other) 4. Was a pathogen detected? (yes, no) if yes: 5. The following pathogen(s) was/were detected. (Appendix A)

**Table 5 animals-14-03346-t005:** Questions: communication frequencies and topics of veterinary communication. (multiple answers were possible: 3. and 4.) [73].

1. The veterinarian spoke with me in this context about zoonoses. (yes, no) 2. The veterinarian spoke with me in this context about AMR. (yes, no) 3. The veterinarian spoke with me about following pathogen(s). (Appendix A) 4. The veterinarian spoke with me about the following topics. (vaccination, treatment of endo- and ectoparasites, handling of animal bites, risk associated feeding practices ^1^, hygiene measures ^2^)

^1^ Feeding raw meat/fish, fresh offal, uncooked bones. ^2^ Picking up dog waste, hand hygiene, cleaning of food and water bowls, washing textiles, associated risks for pregnant people when cleaning the cat litter, birth hygiene, covering of children’s sandpit, cleaning vegetables from the garden, and other.

**Table 6 animals-14-03346-t006:** Questions: perception of veterinarian communication. [73].

Level of Communication [86]	Items (Specific)	Items (Routine)
factual level	The veterinarian explained the cause and transmission of my pet’s infection (e.g., pathogen characteristics).	The veterinarian explained the cause and transmission of zoonoses/AMR (e.g., pathogen characteristics).
factual level	I was able to follow the veterinarian’s explanations on the cause and transmission of my pet’s infection (e.g., language, choice of words, pace of speech).	I was able to follow the veterinarian’s explanations on the cause and transmission of zoonoses/AMR (e.g., language, choice of words, pace of speech).
factual level	The veterinarian’s explanations of the cause and transmission of my pet’s infection were formulated in a language I could understand.	The veterinarian’s explanations of the cause and transmission of zoonoses/AMR were formulated in a language I could understand.
factual level	The veterinarian emphasised the most important points about the cause and transmission of my pet’s infection (e.g., transmission mechanism, pathogen characteristics).	The veterinarian emphasised the most important points on the cause and transmission of zoonoses/AMR (e.g., transmission mechanism, pathogen characteristics).
appeal level	The veterinarian provided me with instructions to prevent the transmission of my pet’s infection (e.g., cleaning and disinfecting hands, food bowls, toys, pet beds, baskets, blankets and pet toilets).	The veterinarian provided me with instructions to prevent the transmission of zoonoses/AMR (e.g., cleaning and disinfecting hands, food bowls, toys, pet beds, baskets, blankets and pet toilets).
appeal level	The veterinarian explained the reasons for the recommended behavioural instructions connected to my pet’s infection (e.g., preventing the transmission of the pathogen to me, maintaining the infection and reinfection of the pet).	The veterinarian explained the reasons for the recommended behavioural instructions connected to zoonoses/AMR (e.g., preventing the transmission of the pathogen to me, maintaining the infection and reinfection of the pet).
appeal level	The veterinarian explained to me the advantages of the behavioural instructions connected to my pet’s infection (e.g., prevention of new outbreaks of disease, health care and health protection, cost savings).	The veterinarian explained to me the advantages of the behavioural instructions connected to zoonoses/AMR (e.g., prevention of new outbreaks of disease, health care and health protection, cost savings).
appeal level	The veterinarian provided me with information material on my pet’s infection (e.g., information sheets, magazines, websites).	The veterinarian provided me with information material on zoonoses/AMR (e.g., information sheets, magazines, websites).
relationship level	The veterinarian demonstrated understanding towards me during the conversation on my pet’s infection.	The veterinarian demonstrated understanding towards me during the conversation on zoonoses/AMR.
relationship level	I had the feeling that the veterinarian took my questions serious during the conversation on my pet’s infection.	I had the feeling that the veterinarian took my questions serious during the conversation on zoonoses/AMR.
relationship level	The veterinarian was able to understand what the behavioural instructions, connected to my pet’s infection, mean to me.	The veterinarian was able to understand what the behavioural instructions, connected to zoonoses/AMR, mean to me.
relationship level	I trust the veterinarian’s advice connected to my pet’s infection.	I trust the veterinarian’s advice connected to zoonoses/AMR.
self-disclosure	I felt that the veterinarian gave me competent advice connected to my pet’s infection.	I felt that the veterinarian gave me competent advice connected to zoonoses/AMR.
self-disclosure	I had the feeling that the conversation with the veterinarian on my pet’s infection, took place at eye level.	I had the feeling that the conversation with the veterinarian on zoonoses/AMR took place at eye level.
self-disclosure	I had the feeling that I could ask the veterinarian questions about my pet’s infection at any time.	I had the feeling that I could ask the veterinarian questions about zoonoses/AMR at any time.
self-disclosure	I had the feeling that it was important to the veterinarian that I understand the causes and consequences of my pet’s infection.	I had the feeling that it was important to the veterinarian that I understand the causes and consequences of zoonoses/AMR.

**Table 7 animals-14-03346-t007:** Questions: veterinary advice and needs of companion pet owners. (multiple answers were possible: 3.) [73].

1. Would you like to receive veterinary advice on zoonoses? (yes, no) 2. Would you like to receive veterinary advice on AMR? (yes, no) 3. Would you like to receive veterinary advice on any of the following topics: (nutrition, behaviour, animal husbandry, vaccination, hygiene measures, treatment of endo- and ectoparasites)

## Data Availability

Data are contained within the article or Appendix A.

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
