# Peer review of "A Survey on Companion Animal Owners’ Perception of Veterinarians’ Communication About Zoonoses and Antimicrobial Resistance in Germany"

_animals, 2024, doi:10.3390/ani14223346_

Round 1

Reviewer 1 Report

Comments and Suggestions for Authors

This is an interesting paper which investigates communication between vets and clients regarding zoonotic diseases. This is becoming a larger issue, especially when associated with raw feeding, and a larger number of old people within the population who maybe immunocompromised.

It is well executed and well set up, and as a result, I only have a few very minor comments which I have detailed below.

Line 68- perhaps drug resistant pathogens rather than just resistant pathogen

I struggled to follow section 2.2.5.2, but that may just be because it is not something that I am over familiar with. Please can you just give it a read, and if you are happy with it then no need to change it

Line 214- do you mean the entire questionnaire?

Line 251- was the origin of the imported animal investigated?

Figure 8- is there any way to make the text here larger- it is very difficult to read

Line 398- its likely that GI and skin issues are the major route of transmission of zoonotic pathogens

Line 409-411- and again, these also increase the risk of transmission to humans through accidental indoor soiling or skin contact via petting or through furniture contact

Line 440- Leishmania needs to be capitalised and italicised

I know that the authors include the questions in the relevant parts in the methodology, but it would be nice to see the full questionnaire included in the supplementary data for simplicity and to allow the study to be repeated elsewhere if researchers so desired. Not a requirement, more a thought.

Author Response

Thank you very much for taking the time to review this manuscript. Please find the detailed responses below and the adaptations in the manuscript (word file in editing mode).

Point 1: Line 68- perhaps drug resistant pathogens rather than just resistant pathogen.

Response 1: Many thanks for this valuable suggestion. We've updated the text to ‘antimicrobial-resistant pathogens’ to ensure clarity (line 83).

Point 2: I struggled to follow section 2.2.5.2, but that may just be because it is not something that I am over familiar with. Please can you just give it a read, and if you are happy with it then no need to change it.

Response 2: Thanks a lot for pointing this out and sorry for the confusion. We adapted and restructured the chapter for a better readability (line 212).

Point 3: Line 214- do you mean the entire questionnaire?

Response 3: Yes, we included only participants who finished all chapters of the questionnaire in our analysis. However, because no questions were mandatory, participants could choose to skip any question within each chapter if they wanted. We have now added this detail (line 273), and we really appreciate you bringing it to our attention. Thank you so much for the helpful suggestion!

Point 4: Line 251- was the origin of the imported animal investigated?

Response 4: We only inquired whether the animal had been imported. However, examining the origin could have provided valuable insights. Many organizations that import animals to Germany collaborate with or are based in Southern and Eastern European countries.

Point 5: Figure 8- is there any way to make the text here larger- it is very difficult to read.

Response 5: Thank you for pointing that out! The editors will adjust the size of the figures, so they should be easier to read in the final version. We have made sure to provide high-resolution versions, which will help to make the text larger and clearer. We appreciate your feedback!

Point 6: Line 398- its likely that GI and skin issues are the major route of transmission of zoonotic pathogens

Response 6: Thanks a lot for pointing this out. Yes, we agree. This has also been investigated in other German (#1Health PREVENT) and international research consortia (PET-Risk). We added a sentence according to this (line 496).

Point 7: Line 409-411- and again, these also increase the risk of transmission to humans through accidental indoor soiling or skin contact via petting or through furniture contact.

Response 7: Thank you very much for highlighting this! We completely agree with your point. Similar as above, this has also been investigated in the consortia mentioned in Response 6. We've added a sentence to reflect this important point (line 511).

Point 8: Line 440- Leishmania needs to be capitalised and italicized.

Response 8: Thank you very much! We adapted it to leishmaniosis instead (line 548).

Point 9: I know that the authors include the questions in the relevant parts in the methodology, but it would be nice to see the full questionnaire included in the supplementary data for simplicity and to allow the study to be repeated elsewhere if researchers so desired. Not a requirement, more a thought.

Response 9: Thank you very much for this valuable suggestion. We have carefully considered your point and discussed it further. Ultimately, we decided to incorporate the relevant questions directly within the text rather than in the supplementary data. The original survey is quite extensive, and only a selection of questions was utilized for this study. Therefore, including the entire questionnaire in the supplementary data might introduce unnecessary complexity.

Reviewer 2 Report

Comments and Suggestions for Authors

The study is about a survey on cats and dogs owners’ perception of veterinarians’ communication on zoonoses and antimicrobial resistance.

Line 62. You may add a topic sentence in the start of paragraph instead of starting with however.

Line 64-65: "There are multiple pathogens that can be transmitted from pets to their owners, potentially leading to illness". Kindly add few examples of pathogens.

Line 65-66: "Furthermore, antimicrobial-resistant pathogens pose a health threat, as they can reduce the effectiveness of treatment". Add some examples of AMR pathogens.

Though you have provided list of pathogens in the supplementary data, you should add examples in the introduction to provide sufficient information about pathogens in the introduction and make it reader friendly.

What type of zoonotic pathogens are common in companion animals in Germany? From which area of Germany, pet owners participated more in the survey?

Line 83-84: "Thus, health education, science communication, and knowledge transfer represent essential methods for preventing disease outbreaks and their spread". How can you support your this statement?

Line 101: In addition, the perception of risk varies depending on factors, such as expectations, personal needs, and past experiences. There should be some information or communication on signs and symptoms of illnesses to reduce the risk of zoonotic infections.

Methods are explained well.

Line 348, 355, 364. Are these headings?

Line 404: A total of 114 cases - including bacteria, fungi, protozoa, ectoparasites and endoparasites - were reported. Giardia, Dermatophytes and E. coli were the pathogens detected most frequently. Suddenly authors are discussing pathogens. Kindly add some information in introduction.

Discussion is too long. It will distract the reader.

You may add heading for limitations of study.

Is there any role of creating more awareness in pet owners by workshops, lectures, seminars about recent trends in zoonotic diseases, risk assessments and sign and symptoms of these infections?

Conclusion: Line 510-512: Furthermore, the findings emphasise the importance of routine veterinary care, including deworming, vaccination and hygiene measures for protecting pets and humans from zoonotic diseases and AMR, as numerous pathogens were detected. Check spellings of emphasis. There is no discussion of vaccination in the introduction and discussion, suddenly authors mentioned in the conclusion. Though it has very important role in the risk assessment.

Comments on the Quality of English Language

The manuscript needs English Language editing.

Author Response

Thank you very much for taking the time to review this manuscript. Please find the detailed responses below and the adaptations in the manuscript (word file in editing mode).

Point 1: Line 62. You may add a topic sentence in the start of paragraph instead of starting with however.

Response 1: Thanks a lot for this valuable suggestion. We have changed the beginning of this paragraph (line 66).

Point 2: Line 64-65: ‘There are multiple pathogens that can be transmitted from pets to their owners, potentially leading to illness’. Kindly add few examples of pathogens.

Response 2: Thanks for pointing that out! We added a few examples (line 69).

Point 3: Line 65-66: ‘Furthermore, antimicrobial-resistant pathogens pose a health threat, as they can reduce the effectiveness of treatment’. Add some examples of AMR pathogens.

Response 3: As above, we added a few examples (line 69 and line 77).

Point 4: Though you have provided list of pathogens in the supplementary data, you should add examples in the introduction to provide sufficient information about pathogens in the introduction and make it reader friendly.

Response 4: Thanks a lot for this important point. We added it to the introduction (line 69, 73, 77).

Point 5: What type of zoonotic pathogens are common in companion animals in Germany?

Response 5: In Germany, zoonotic pathogens in companion animals are an emerging concern, but systematic data on their prevalence is limited. This gap is largely since there is no central database specifically for companion animals, and most of these pathogens are not ‘meldepflichtig’ (reportable to health authorities). Without mandatory reporting requirements, cases often go under-reported, leaving only isolated studies or case reports available to understand the prevalence and impact of these zoonotic diseases. In Germany, only a few zoonoses (e.g., campylobacteriosis, echinococcosis, listeriosis, salmonellosis, SARS-CoV2, tuberculosis and verotoxin-producing Escherichia coli infec-tions) currently being considered as reportable animal diseases. We added a paragraph according to this in the introduction (line 73). Thank you for pointing this out.

Point 6: From which area of Germany, pet owners participated more in the survey?

Response 6: Since we did not assess the areas within Germany, we are unable to determine which specific area had more pet owner participation in the survey. The data collected does not include geographical distinctions, so we cannot provide insights into regional variations in participation. For a future survey, including a question about location within Germany would allow us to analyze regional trends in pet ownership and participation rates. Thanks a lot for bringing this up. We added a sentence according to this in the limitations and future perspectives section (line 616).

Point 7: Line 83-84: ‘Thus, health education, science communication, and knowledge transfer represent essential methods for preventing disease outbreaks and their spread’. How can you support your this statement?

Response 7: We added two more sources and adjusted the sentence (line 100) to draw a link to our study. Thank you for your thoughtful feedback.

Point 8: Line 101: In addition, the perception of risk varies depending on factors, such as expectations, personal needs, and past experiences. There should be some information or communication on signs and symptoms of illnesses to reduce the risk of zoonotic infections.

Response 8: Thank you for your thoughtful feedback. Yes, we agree on the importance. That’s why we also included questions about the symptoms to see if there is a variation in communication in comparison to routine consultations. Routine communication is important as a base and to create an understanding. But targeted communication - especially in the case of sickness - is needed to directly prevent pathogen transmission. We added a sentence accordingly (line 113) and in the future perspectives (line 622). Thank you for pointing this out.

Point 9: Methods are explained well.

Response 9: Thank you!

Point 10: Line 348, 355, 364. Are these headings?

Response 9: Yes, the first sentence below the figures serves as the heading. I added a period to make it clearer. Thank you for pointing this out.

Point 11: Line 404: A total of 114 cases - including bacteria, fungi, protozoa, ectoparasites and endoparasites - were reported. Giardia, Dermatophytes and E. coli were the pathogens detected most frequently. Suddenly authors are discussing pathogens. Kindly add some information in introduction.

Response 11: Many thanks for this valuable suggestion. Following your earlier advice, we have now included pathogens in the introduction (line 69, 73, 77).

Point 12: Discussion is too long. It will distract the reader.

Response 12: The length of the discussion is in agreement with the journal’s policy. We modified a few sentences to make the discussion a bit more concise and better readable. We hope that you will agree to this as well.

Point 13: You may add heading for limitations of study.

Response 13: Thank you for the suggestion! The layout we’re using follows the guidelines set by the journal ‘Animals’, which does not allow for adding an additional heading specifically for the study’s limitations. However, we’ve made sure to clearly discuss them within the existing structure and used a separate paragraph for the limitations at the end of the discussion.

Point 14: Is there any role of creating more awareness in pet owners by workshops, lectures, seminars about recent trends in zoonotic diseases, risk assessments and sign and symptoms of these infections?

Response 14: Thank you for bringing up such an important point! We completely agree that creating awareness among pet owners through workshops, lectures, and seminars could be highly effective. In our study, we focused on the role that veterinarians play in risk communication during consultations, but we recognize that additional educational efforts led by professionals would be incredibly valuable. Starting with veterinarians sharing information and providing resources is a great first step, and expanding this to broader awareness campaigns could have a significant impact on public understanding of zoonotic diseases. By the way, we have started in 2023 a lecture series for the public at our Veterinary Centre for Resistance Research at Freie Universität Berlin, in which we deliver basic knowledge on antimicrobial and antiparasitic resistance to animal owners, mainly pet animal owners. We added a part on future perspectives (line 622) in the discussion section.

Point 15: Conclusion: Line 510-512: Furthermore, the findings emphasise the importance of routine veterinary care, including deworming, vaccination and hygiene measures for protecting pets and humans from zoonotic diseases and AMR, as numerous pathogens were detected. Check spellings of emphasis.

Response 15: Changed to ‘emphasize’ (line 635). Thanks a lot!

Point 16: There is no discussion of vaccination in the introduction and discussion, suddenly authors mentioned in the conclusion. Though it has very important role in the risk assessment.

Response 16: Thank you for this advice. We counted vaccination as a protective measure and did not mention it in particular. We now added it in the introduction and discussion (line 93, 554). Moreover, we have replaced immunization by vaccination to stick to one term.

Point 17: Language- The English could be improved to more clearly express the research.

Response 17: Thank you for your feedback regarding the clarity of the language. To ensure the highest quality, we engaged a native English speaker (Dr. Karsten Tedin) with a background in academic writing to thoroughly review and refine the text. We focused on enhancing readability and clarity to better convey our research findings. We hope these improvements meet your expectations.
